# Statistical Methods for Multi-Omics Analysis in Neurodevelopmental Disorders: From High Dimensionality to Mechanistic Insight

**DOI:** 10.3390/biom15101401

**Published:** 2025-10-02

**Authors:** Manuel Airoldi, Veronica Remori, Mauro Fasano

**Affiliations:** 1Department of Science and High Technology, University of Insubria, 22100 Como, Italy; mairoldi@uninsubria.it (M.A.); vremori@uninsubria.it (V.R.); 2Center of Neuroscience, University of Insubria, 21052 Busto Arsizio, Italy

**Keywords:** neurodevelopmental disorders, multi-omics integration, study design, wide data

## Abstract

Neurodevelopmental disorders (NDDs), including autism spectrum disorder, intellectual disability, and attention-deficit/hyperactivity disorder, are genetically and phenotypically heterogeneous conditions affecting millions worldwide. High-throughput omics technologies—transcriptomics, proteomics, metabolomics, and epigenomics—offer a unique opportunity to link genetic variation to molecular and cellular mechanisms underlying these disorders. However, the high dimensionality, sparsity, batch effects, and complex covariance structures of omics data present significant statistical challenges, requiring robust normalization, batch correction, imputation, dimensionality reduction, and multivariate modeling approaches. This review provides a comprehensive overview of statistical frameworks for analyzing high-dimensional omics datasets in NDDs, including univariate and multivariate models, penalized regression, sparse canonical correlation analysis, partial least squares, and integrative multi-omics methods such as DIABLO, similarity network fusion, and MOFA. We illustrate how these approaches have revealed convergent molecular signatures—synaptic, mitochondrial, and immune dysregulation—across transcriptomic, proteomic, and metabolomic layers in human cohorts and experimental models. Finally, we discuss emerging strategies, including single-cell and spatially resolved omics, machine learning-driven integration, and longitudinal multi-modal analyses, highlighting their potential to translate complex molecular patterns into mechanistic insights, biomarkers, and therapeutic targets. Integrative multi-omics analyses, grounded in rigorous statistical methodology, are poised to advance mechanistic understanding and precision medicine in NDDs.

## 1. Introduction

Neurodevelopmental disorders (NDDs), including autism spectrum disorder (ASD), intellectual disability (ID), attention-deficit/hyperactivity disorder (ADHD), Rett syndrome, and CDKL5 deficiency disorder, represent a diverse group of conditions that affect approximately 1–3% of children worldwide. Meta-analyses estimate ASD prevalence at ~0.7–1% globally [1,2], ID around 2% [3], and ADHD near 10% in boys and 5% in girls [4]. Rett syndrome and CDKL5 deficiency disorder are much rarer, with estimated prevalences of ~1 in 10,000–15,000 and ~1 in 40,000–60,000 live births, respectively, and are typically caused by highly penetrant monogenic mutations [5,6]. Despite their rarity, these monogenic disorders have provided critical mechanistic insights into neurodevelopmental pathways and synaptic function, serving as model systems to understand complex NDDs.

The genetic architecture of NDDs is highly heterogeneous, encompassing rare, high-penetrance variants (e.g., de novo single-nucleotide variants and copy number variants) as well as the cumulative effects of common alleles contributing to polygenic risk [7,8]. Large-scale sequencing studies, including whole-exome sequencing (WES) and whole-genome sequencing (WGS), have substantially increased diagnostic yields, reaching 20–40% in trio-based studies. Emerging long-read sequencing technologies and integrative approaches that combine WES/WGS with transcriptomic, epigenomic, or methylome profiling are further enhancing diagnostic resolution [9,10]. Nevertheless, a substantial gap remains between variant detection and functional or mechanistic interpretation, which underscores the necessity of complementary molecular and computational strategies [11].

High-throughput omics technologies provide complementary perspectives on these mechanisms. Transcriptomic profiling captures gene expression dysregulation, alternative splicing, and allele-specific expression in NDDs [12], while proteomics quantifies protein abundance, post-translational modifications (PTMs) such as phosphorylation or ubiquitination, and protein–protein interactions, providing functional insights that are not always inferable from RNA-level data [3,4]. Correlations between mRNA and protein levels in human brain tissue are often modest, reflecting complex post-transcriptional regulation, protein turnover, and tissue-specific translational control [5]. Structural proteomics, phosphoproteomics, and interactomics are increasingly leveraged to understand synaptic function, neuroplasticity, and signaling pathway alterations in ASD and related NDDs [13]. Integrating these molecular layers offers the potential to bridge genetic variation with cellular phenotypes and disease-relevant pathways.

Beyond single-layer analyses, the integration of multiple molecular layers—genomic, transcriptomic, proteomic, metabolomic, and epigenomic— provides the opportunity to bridge genetic variation with cellular phenotypes, signaling pathways, and disease-relevant networks. Such multi-omics integration is particularly valuable in NDDs, where perturbations are often subtle, distributed across interconnected pathways, and context-dependent. For example, a rare de novo mutation in a synaptic gene may trigger downstream dysregulation of both protein networks and metabolic pathways, which can only be captured through integrative approaches.

However, omics studies in NDDs face persistent statistical and computational challenges. High dimensionality, batch effects, sparsity, and multiple testing burdens complicate the analysis of genome-scale data. Proper normalization and variance modeling are critical to distinguish biological signal from technical noise. For instance, DESeq2’s median-of-ratios approach addresses library size variability in RNA-seq [6], whereas proteomics datasets often rely on quantile normalization, internal reference standards, or vendor-specific algorithms to mitigate technical artifacts. Quality control (QC) steps, which include assessment of sample integrity, detection of technical outliers, and evaluation of dataset-wide metrics such as mapping rates, duplication levels, or signal-to-noise ratios, are equally important. Indeed, poor QC can severely compromise downstream inference, introducing artifacts that persist even after normalization and batch correction. For example, outlier samples due to RNA degradation or low sequencing depth can distort differential expression analyses and bias integrative modeling. In neurodevelopmental disorder studies, where phenotypic and molecular variability is already high, failure to identify and exclude low-quality data can exacerbate false discoveries or obscure true biological signals [14,15]. Additionally, the adjustment for latent or known confounders, such as age, sex, brain region, or technical covariates, is critical, particularly in NDD studies where case–control imbalances or developmental stage effects are common [16]. Downstream, shrinkage estimation, penalized regression, and feature filtering are necessary to extract robust, biologically meaningful signals from high-dimensional datasets [7,8,9]. These challenges are amplified in multi-omics integration, where heterogeneous data types and differing levels of missingness necessitate careful modeling. Importantly, the complexity of NDDs also arises from phenotypic heterogeneity and overlapping comorbidities. Patients may present with combinations of cognitive, behavioral, and motor deficits, often influenced by environmental factors and developmental stages. This heterogeneity complicates cohort selection, study design, and the generalizability of findings. Moreover, sex differences and ancestry-specific genetic effects can contribute to differential risk and molecular phenotypes, emphasizing the need for stratified analyses and careful consideration of confounding factors.

Overall, advances in high-throughput omics technologies, together with sophisticated computational methods, have opened unprecedented opportunities to dissect the molecular architecture of NDDs. Integrating genomic, transcriptomic, and proteomic data allows for the mapping of disease-associated variants to functional consequences, regulatory networks, and cellular phenotypes. By leveraging these approaches, researchers can identify convergent molecular pathways, propose candidate biomarkers, and ultimately inform precision therapeutic strategies for diverse NDD populations.

## 2. Statistical Challenges in High-Dimensional Omics Data

High-throughput omics platforms generate so-called wide data, characterized by thousands of features measured in relatively small sample cohorts. This “large p, small n” scenario (where the number of features greatly exceeds the number of samples) increases the risk of overfitting, spurious associations, and irreproducible findings if not properly managed [17,18]. This imbalance complicates traditional statistical inference because standard methods assume that the number of observations exceeds the number of variables, a condition violated in typical omics datasets. Consequently, specialized statistical frameworks that explicitly model noise, dependence structures, and sparsity are necessary to ensure robust inference and reproducibility.

Data preprocessing procedures, such as normalization, are critical first steps to mitigate technical artifacts, addressing biases such as library size variability in RNA-seq or labeling and ionization differences in mass spectrometry-based proteomics. Common transcriptomic normalization methods include the median-of-ratios implemented in DESeq2 [19], trimmed mean of M values (TMM) from edgeR [20] and quantile normalization [21]. Proteomics normalization often relies on quantile scaling, internal reference standards, or variance-stabilizing normalization [22]. Failure to appropriately normalize data can result in confounding technical variation with biological differences, leading to false conclusions. Recent advances also include methods such as RUVSeq (Remove Unwanted Variation) that leverage control genes or samples to improve normalization accuracy [23]. Normalization strategies must be tailored to the omics platform and experimental design; no one-size-fits-all approach exists.

Batch effects and hidden confounders constitute another major challenge. Differences in sample handling, reagents, instrumentation, or even operator can introduce systematic noise that obscures true biological signals [24,25]. Surrogate variable analysis (SVA) [26] and factor-based methods [27] are widely applied in transcriptomics. In contrast, ComBat [24] and Limma’s *removeBatchEffect()* [28] are widely used in proteomics, although they were originally designed for transcriptomic data. These methods aim to preserve biological heterogeneity while mitigating technical artifacts, though overcorrection can inadvertently remove relevant signals. In addition, emerging approaches such as harmonization via mutual nearest neighbors (MNN) and deep learning-based batch correction algorithms are gaining traction for their ability to handle complex batch structures, especially in single-cell omics [29,30]. In NDD studies, batch correction is particularly critical when combining data across brain regions, developmental stages, or experimental models (e.g., cerebral organoids, iPSC-derived neurons), which may introduce subtle but biologically meaningful variance that can be mistaken for noise.

Cohort heterogeneity adds another layer of complexity. Differences in sex, age, ancestry, disease severity, comorbidities, and medication status can all influence molecular measurements, introducing variance that is not disease-related [31,32]. Study design factors, including sampling strategies, tissue type, postmortem interval, and developmental stage, further introduce variance that may obscure true disease-associated signals [33]. Longitudinal and repeated-measures designs help mitigate some of these challenges by capturing intra-individual variability over time, thereby improving statistical power to detect disease-relevant changes [34]. Integrative analyses of single-cell and spatially resolved omics can deconvolve mixed cell populations, revealing cell-type-specific effects that are otherwise hidden in bulk measurements [35]. Computational frameworks that model heterogeneity explicitly, including mixed-effects models, Bayesian hierarchical approaches, and matrix factorization methods, have been shown to improve robustness and reproducibility in NDD studies [26]. These models not only account for known sources of variability but also allow for the discovery of latent structures in the data, such as patient subgroups or tissue-specific modules, that may correspond to distinct disease mechanisms or developmental trajectories. Figure 1 provides an overview of key sources of biological and technical variability that should be considered when designing multi-omics studies of neurodevelopmental disorders, including cohort characteristics, sample-related factors, and study design structure. Addressing cohort heterogeneity and study design confounders is particularly important when datasets are integrated across multiple sites, experimental platforms, or omics layers. Failure to account for these factors can lead to biased conclusions, inflated false positives, and poor generalizability of findings, undermining the potential of integrative multi-omics approaches.

Missing values and zero inflation are particularly problematic in proteomics datasets acquired via data-dependent acquisition (DDA), where observed zeros may reflect stochastic sampling rather than true absence of expression. While the optimal solution to the missing values problem is to prevent them during data collection, by optimizing sample preparation, increasing analytical depth, or using data-independent acquisition methods, this is not always feasible, particularly in clinical or resource-limited settings. Thus, several strategies are commonly employed when this is not feasible. Listwise deletion, one of the most common procedures, involves excluding observations containing missing values, but this approach may remove a large fraction of the original data [36]. Alternatively, imputation strategies, including K-nearest neighbors, random forest-based approaches, or left-censored minimal value imputation help preserve variance structure [37]. However, each method presents specific trade-offs: K-nearest neighbors and random forest imputation can preserve local data structure but may introduce bias when missingness is not random, while minimal value imputation risks underestimating true variability and inflating false positives. Poorly chosen imputation strategies can distort the underlying variance structure of the dataset, leading to misleading clustering, spurious associations, or attenuation of biological signals. Notably, recent advances in probabilistic modeling and deep learning, such as Bayesian missing data models and autoencoder-based imputations, offer promising avenues to recover missing proteomic measurements while retaining biological variability [38,39]. Yet, these methods also require careful parameter tuning and validation, and their assumptions may not be appropriate for all omics contexts. A practical approach should therefore involve careful assessment of the missing data mechanism and the specific characteristics of the omics layer being analyzed.

The multiple testing burden is another central challenge. Tens of thousands of features are typically analyzed simultaneously, necessitating control of false positive occurrences. There is a number of existing methods addressing this problem, and one of the most commonly used and simple approach is the Bonferroni correction, which relies on the Family Wise Error Rate (FWER). The Bonferroni method controls FWER by computing the adjusted *p*-values, multiplying the original *p*-values by the number of tested hypotheses. A drawback of this procedure is that it is extremely conservative. The Benjamini–Hochberg procedure [40] increases the method’s power by controlling the false discovery rate (FDR), with advanced approaches such as shrinkage-based estimators and penalized regression methods (e.g., LASSO, Elastic Net, Priority-Elastic Net) improving power while controlling false positives [41]. In addition, hierarchical testing frameworks and adaptive FDR methods have been developed to exploit the dependency structure of omics data, further enhancing detection power [42,43].

Dimensionality reduction and the management of multicollinearity are essential steps to facilitate both pattern discovery and predictive modeling in high-dimensional biological data. As said before, large scale omics datasets contained thousands of correlated features, which can obscure meaningful biological signals and compromise statistical inference. Matrix factorization methods, such as principal component analysis (PCA) [44] and independent component analysis (ICA) [45], provide a general framework to address these problems by decomposing the original dataset into a reduced set of latent variables that capture the most relevant sources of variation. Closely related latent variable methods, including partial least squares discriminant analysis (PLS-DA), and sparse canonical correlation analysis (sCCA) [46,47], extend this principle by incorporating group information or integrating multiple datasets. Collectively, these methods provide a powerful toolkit to summarize high-dimensional datasets while retaining key biological variation. These approaches reduce overfitting, enhance interpretability, and provide a foundation for integrative multi-omics analyses. Moreover, nonlinear methods such as t-distributed stochastic neighbor embedding (t-SNE) and Uniform Manifold Approximation and Projection (UMAP) have become popular for visualizing complex omics data in lower dimensions, facilitating the identification of subtle biological subgroups [48].

Careful attention to normalization, batch correction, missing data handling, multiple testing, and dimensionality reduction forms the backbone of robust statistical frameworks for high-dimensional omics studies [17,19,25,28], essential before applying analyses to complex biological contexts such as neurodevelopmental disorders. Equally important is the validation of findings across independent datasets or cohorts, which can strengthen biological conclusions and mitigate overfitting. Validation, benchmarking, and reproducibility are critical components to ensure the reliability of multi-omics findings. Independent dataset validation, either through external cohorts or cross-validation strategies, helps confirm that identified biomarkers or molecular signatures are robust and generalizable across different populations and technical conditions [49]. Benchmarking studies comparing different statistical methods and integration pipelines provide valuable insights into their relative performance, strengths, and limitations, guiding appropriate method selection for specific research questions [50]. Furthermore, reproducibility is challenged by variations in data preprocessing, batch effects, and analytical choices. Thus, transparent reporting, standardized workflows, and availability of code and data are essential to enable reproducibility and facilitate cumulative knowledge building in the NDD field. As omics data continue to grow in volume and complexity, ongoing methodological developments will be critical to unlock their full potential in deciphering disease mechanisms and guiding precision medicine.

## 3. Univariate vs. Multivariate Models in Transcriptomics and Proteomics

Omics datasets in NDDs are often high-dimensional, with thousands of genes, proteins, or metabolites measured in relatively small cohorts. These characteristics pose unique statistical and computational challenges. Selecting the appropriate analytical strategy is critical to avoid spurious associations, control overfitting, and maximize biological insight. Univariate approaches examine each feature independently to test for differences across conditions. They are computationally efficient, straightforward to interpret, and well-suited for hypothesis-driven analyses. In transcriptomics, commonly used tools include DESeq2 [19], edgeR [28], and Limma [28], which model gene expression data under assumptions appropriate for count-based or normalized data and employ robust variance estimation and empirical Bayes shrinkage. The choice between DESeq2, edgeR, and Limma depends on the specific characteristics of the dataset and the assumptions underlying each method. DESeq2 and edgeR both rely on negative binomial models but differ in their approaches to normalization and dispersion estimation. DESeq2 applies a median-of-ratios normalization and uses more conservative dispersion estimates, making it particularly suitable for datasets with moderate-to-high variance and small sample sizes, conditions frequently encountered in NDD studies. edgeR, while also robust, tends to perform better with larger sample sizes or higher counts per feature due to its more flexible estimation of dispersion. Limma, originally developed for microarray data and extended to RNA-seq through the voom transformation, is effective when working with normalized expression values and leverages empirical Bayes shrinkage to stabilize variance estimates across features, often performing well in small-N scenarios. In NDD studies, where cohorts are often limited, heterogeneous, or noisy, DESeq2 and Limma are commonly preferred for their stability and capacity to handle heteroscedasticity. Recent studies illustrate these methodological differences in practical contexts. In a transcriptomic analysis of human neural progenitors exposed to low-grade environmental factors linked to autism, DESeq2 identified convergent upregulation of synaptic and lipid metabolism pathways, demonstrating its sensitivity in small-scale, exposure-based experimental designs [51]. edgeR, alongside DESeq2, was used in a study of patient-derived retinal organoids from individuals with optic nerve hypoplasia (ONH), where both methods revealed overlapping sets of differentially expressed genes enriched for known NDD-related pathways, supporting the genetic contribution to ONH [52]. Limma was applied in an integrated microarray analysis to construct an immune-related diagnostic model for ASD, identifying 41 differentially expressed genes and implicating specific immune cell populations such as neutrophils and CD8+ T cells [53]. These examples highlight how tool selection can influence both the sensitivity and interpretability of findings in NDD research.

In proteomics, classical statistical tests, such as Student’s *t*-tests, ANOVA, and their non-parametric equivalents, are widely applied [17]. Univariate tests are straightforward to interpret and robust under multiple testing correction (e.g., Benjamini–Hochberg [40]). For example, Limma and Student’s *t*-test have been used to identify differentially expressed proteins in extracellular vesicles from plasma of individuals with ASD, revealing five downregulated proteins, including CD40 and HSP27, with potential as diagnostic biomarkers [54]. One-way ANOVA and Tukey’s post hoc tests were applied to quantify spatiotemporal protein expression changes in dopaminergic neurons across developmental stages, capturing molecular signatures relevant to neuronal maturation [55]. These examples illustrate how univariate frameworks remain central in proteomic studies of NDDs, particularly when combined with proper normalization and multiple testing correction.

Furthermore, in NDD studies, univariate analyses have successfully identified disease-associated molecular changes. For example, differential gene expression analyses in postmortem ASD brain tissue revealed alterations in synaptic and immune-related pathways [8]. In Rett syndrome, univariate analyses of proteomic datasets highlighted altered abundances of proteins involved in chromatin remodeling and neuronal development, providing initial insights into molecular dysregulation [12]. However, univariate methods inherently overlook inter-feature dependencies and systems-level organization. Many molecular pathways and regulatory networks involve coordinated changes across multiple genes, proteins, or metabolites; univariate tests, by evaluating each feature in isolation, may miss these subtle, multicomponent alterations.

Multivariate approaches capture the joint behavior of multiple features, allowing the identification of covariance structures, latent variables, and correlated patterns that reflect systems-level organization. Dimensionality reduction techniques such as PCA [44], ICA [45], and PLS-DA [46] reduce dimensionality while highlighting major sources of variation. Sparse extensions, including sPLS-DA and sCCA [47,56,57], combine feature selection with dimensionality reduction, improving interpretability and reducing overfitting in small cohorts.

In NDD research, multivariate models have revealed coordinated dysregulation across multiple molecular layers. For instance, studies integrating transcriptomics and proteomics in ASD models identified simultaneous disruptions in synaptic proteins and mitochondrial pathways [58]. Moreover, integrative analyses combining metabolomics with transcriptomics uncovered immune-metabolic dysregulation mediated by transcription factors such as RARA and NFKB2 [59]. Penalized regression models, such as LASSO, Elastic Net, and Priority-Elastic Net [41,60], further enhance predictive modeling by identifying informative features while controlling overfitting. These approaches have been successfully applied to multi-omics data from patient-derived cerebral organoids and iPSC-derived neuronal cultures, providing mechanistic insight into convergent molecular signatures across NDDs [58].

Hybrid approaches are increasingly adopted to combine the strengths of univariate and multivariate methods. In a union strategy, features identified by either approach are retained to maximize sensitivity, particularly useful when effect sizes are small or heterogeneous [61,62]. Conversely, intersection strategies, where only features detected by both univariate and multivariate models are selected, enhance specificity, reducing false positives and emphasizing robust, reproducible signals [63]. This integrative selection framework is particularly valuable in NDD research, where disease-associated molecular changes may be subtle, context-dependent, and spread across multiple omics layers. Additionally, combining feature-level statistics from univariate models with latent factors derived from multivariate models (e.g., using multivariate scores as covariates in univariate regression) can further refine associations and improve interpretability [64].

Overall, the choice between univariate and multivariate methods should be guided by the aim of research, data characteristics, and sample size. While univariate tests remain essential for detecting strong, individual-level effects, multivariate and penalized models are critical for uncovering complex, systems-level dysregulation typical of NDDs. In practice, combining both strategies offers a flexible and powerful analytic toolkit capable of adapting to the inherent complexity of multi-omics data in NDDs.

Table 1 provides a summary of the main statistical and integrative methods commonly applied in multi-omics analyses of neurodevelopmental disorders.

## 4. Integrative Multi-Omics Approaches in Neurodevelopmental Disorders

Single-omics analyses, while informative, often fail to capture the full complexity of NDDs, where molecular perturbations span multiple regulatory layers. Integrative multi-omics approaches—combining transcriptomic, proteomic, metabolomic, epigenomic, and other datasets—provide a systems-level perspective that can reveal convergent disease mechanisms and identify candidate biomarkers [1,12,58,59,68]. By linking genetic variation to downstream molecular consequences, these approaches help bridge the gap between variant discovery and functional interpretation [7,8].

Integration strategies vary according to the timing and type of data combination. Early integration involves merging normalized datasets from multiple omics layers prior to statistical modeling. Techniques such as sparse canonical correlation analysis (sCCA) [47] and multi-block partial least squares (MB-PLS) [69] identify latent variables that capture covariation across data types, enabling the detection of molecular patterns invisible in single-layer analyses [46,65]. For instance, a recent work [70] demonstrates that a multi-block sPLS-DA (a discriminant version of MB-PLS with lasso penalization) can be effectively applied to multi-omics data integration. Building on this approach, they combined Polygenic Risk Scores, epigenomics and metabolomics to investigate ADHD. This integration revealed previously reported associations with ADHD, including those related to *MAD1L1* gene and glucocorticoid associations, while also highlighting *STAP2* as a possible novel gene involved in this neurodevelopmental condition.

Supervised frameworks, including DIABLO, further allow integration of omics profiles with clinical or phenotypic outcomes, facilitating the identification of molecular drivers associated with ASD, ID, or ADHD [65]. The main advantage of early integration lies in its ability to uncover coordinated molecular signals across data types, improving statistical power and biological interpretability when datasets are well matched and normalized. However, it requires rigorous preprocessing to harmonize data scales and distributions, and it can be computationally intensive. Additionally, early integration may be less robust in the presence of missing data or heterogeneous sample sets, which are common challenges in NDD research. In contrast, late integration approaches combine results from independent analyses (e.g., intersecting differentially expressed genes with differentially abundant proteins) to highlight convergent pathways and regulatory networks [71]. Also, late integration approaches include pathway and gene set enrichment analyses that identify shared biological processes, network-based methods that integrate multi-omics data through protein–protein interaction or co-expression networks, and meta-dimensional approaches that combine independent omics results while preserving layer-specific information. Such methods allow for flexible, hypothesis-driven interpretation and can better accommodate heterogeneous datasets typical of NDD studies [72,73]. The strengths of late integration include its flexibility to incorporate diverse data types analyzed independently and its relative robustness to missing or unbalanced data. It also facilitates biological interpretation by focusing on convergent signals at the pathway or network level. However, this approach may overlook subtle, cross-omics interactions detectable only through joint modeling, and independent analyses may differ in statistical power or bias, complicating integration. By understanding the advantages and limitations of each strategy, researchers can select the most appropriate integration approach based on study design, data characteristics, and the specific hypotheses being tested.

Several studies illustrate the utility of these integrative strategies in NDD contexts. In a *Cntnap2* knockout mouse model and human cerebral organoids derived from individuals with ASD, combined transcriptomic and proteomic profiling revealed consistent alterations in synaptic function, mitochondrial activity, and axonal architecture, particularly in excitatory neurons [58]. Urine proteomics and metabolomics in children with ASD identified co-regulated changes in glutathione metabolism, xenobiotic detoxification, and immune-related pathways, suggesting non-invasive biomarkers of neuroinflammation [68]. Similarly, combined transcriptomics and metabolomics analyses demonstrated immune activation, synaptic dysfunction, and metabolic dysregulation in ASD, with transcription factors such as RARA and NFKB2 modulating these interconnected pathways [59].

A recent multi-omics study integrating metagenomics, metaproteomics, host proteomics, and metabolomics from fecal samples of children with ASD provided a comprehensive view of microbiome–host interactions. Using a customized bacterial protein database based on 16S rRNA sequencing, combined with robust normalization and statistical frameworks, this approach linked microbial and metabolic alterations to potential pathophysiological mechanisms in ASD, highlighting the value of integrative multi-omics in unraveling complex neurodevelopmental disorders [72]. The study underscored how integrating host and microbial layers can help disentangle the role of gut–brain axis dysfunctions in ASD, a crucial area given the emerging evidence of microbiota influence on neurodevelopment and behavior. Another study exemplified this by applying transcriptome-wide association modeling (FUSION), summary-based Mendelian randomization (SMR), and Bayesian colocalization (COLOC) to integrate m^6^A-QTLs with GWAS, eQTL, and pQTL datasets, enabling multi-layered inference of regulatory mechanisms across neuropsychiatric disorders, including ASD [74].

Moreover, late multi-omics integration strategies are able to highlight mechanistic insights as demonstrated by recent studies. For instance, in [75] it is evident how late integration of bulk transcriptomics, proteomics and scRNA-seq data has deepened our knowledge of immune pathway alterations in ASD, revealing specific dysregulation in TRAIL, RANKL, and TWEAK pathways specifically in circulating NK cells and T cell subsets rather than general immune pathway dysfunction. Computational strategies are central to robust multi-omics integration. Penalized regression approaches, including Elastic Net and Priority-Elastic Net, are widely applied to high-dimensional datasets for feature selection and predictive modeling, mitigating overfitting while retaining biologically informative signals [41,76]. Similarity network fusion (SNF) aggregates patient similarity matrices across omics layers, enabling molecular stratification of heterogeneous cohorts [66]. Network diffusion models, such as Markov Affinity-based Proteogenomic Signal Diffusion (MAPSD), propagate signals through protein–protein interaction networks, identifying brain-region-specific subnetworks and novel disease-associated genes [77]. Bayesian latent variable models, including Multi-Omics Factor Analysis (MOFA) and DIABLO, allow for missing data handling, uncover latent molecular factors, and provide interpretable links between omics modalities and phenotypes [65,67].

Despite these advances, challenges remain, including differences in data dimensionality, technical noise across platforms, and the need for interpretable models suited to limited sample sizes common in NDD cohorts. Additional hurdles include batch effects, heterogeneity in sample types (e.g., brain region, developmental stage), and the integration of longitudinal multi-omics data, which require sophisticated normalization and harmonization techniques to ensure robust inference. Collectively, these integrative approaches have begun to uncover shared molecular signatures in NDDs, including synaptic imbalance, mitochondrial dysfunction, and immune dysregulation, despite extensive genetic heterogeneity [58,59,68]. The success of multi-omics studies depends on careful cohort design, standardized preprocessing, and independent validation [9].

Furthermore, machine learning strategies have been increasingly applied to integrate heterogeneous datasets, linking clinical phenotypes with molecular profiles to identify robust biomarkers and disease subtypes. For example, combining detailed clinical data from the Autism Diagnostic Interview-Revised (ADI-R) with transcriptomic profiles via supervised and unsupervised machine learning methods has enabled the delineation of ASD subgroups characterized by distinct behavioral and gene expression patterns, thus illustrating the power of integrative computational approaches in neurodevelopmental disorders [78]. Deep learning frameworks and ensemble methods have also been explored to capture complex nonlinear relationships and interactions across omics layers, although their interpretability remains a key challenge that researchers are addressing through methods like SHAP values and attention mechanisms.

Indeed, future directions include the incorporation of spatially resolved and cell-type-specific omics, as well as machine learning strategies, to translate complex molecular patterns into mechanistic insights and potential biomarkers for precision medicine in neurodevelopmental disorders [38]. For instance, recent advances in single-cell multimodal omics technologies enabled simultaneous profiling of multiple molecular layers (e.g., genome, epigenome, transcriptome, proteome) within individual cells, offering unprecedented resolution to dissect cellular heterogeneity and gene regulatory networks in complex tissues such as the brain [79]. Such high-resolution approaches are particularly promising in disorders like ASD and Rett syndrome, where altered neurodevelopment may affect only specific cell types or circuits. These approaches facilitate a comprehensive understanding of cell-type-specific mechanisms underlying neurodevelopmental disorders and hold promises for identifying novel therapeutic targets. Moreover, emerging spatial transcriptomics and proteomics platforms are poised to provide critical insights into the spatial context of molecular changes, allowing researchers to map pathological signatures directly onto brain architecture, which is especially relevant given the region-specific pathology observed in many NDDs.

## 5. Future Directions and Translational Perspectives in Neurodevelopmental Disorders

Despite substantial progress in omics and integrative analyses, several challenges remain in translating multi-omics findings into clinical applications for NDDs. One major limitation is the genetic and phenotypic heterogeneity of NDD cohorts, which hinders the identification of reproducible biomarkers and therapeutic targets [80]. This variability extends not only across patient populations but also across tissues, developmental stages, and even experimental platforms. Future efforts will benefit from larger, deeply phenotyped cohorts combined with harmonized multi-omics pipelines, thereby improving statistical power and mechanistic insights [81]. Longitudinal study designs—where patients are followed across multiple developmental time points—are likely to be particularly powerful in capturing disease trajectories.

ASD exemplifies the complexity of NDDs, encompassing a wide array of genetic, molecular, and behavioral characteristics. Multi-omics approaches are increasingly elucidating the mechanistic underpinnings of ASD. For instance, recent studies have identified novel SHANK2 variants, highlighting disruptions in synaptic genes as contributors to ASD pathophysiology [82]. Additionally, research into neuroepitranscriptomics has revealed that RNA modifications such as m6A and m3C influence neuronal development and synaptic function, linking dysregulated RNA modification to neurodevelopmental deficits [83]. Furthermore, Granger Causality Analysis applied to functional imaging data has uncovered reduced connectivity in the medial prefrontal cortex and amygdala in children with ASD, associating neural network alterations with social cognition deficits and suggesting the potential of imaging biomarkers for diagnosis and longitudinal monitoring [84]. Moreover, AI-based literature mining systems have demonstrated the utility of computational approaches in integrating genomics, transcriptomics, and proteomics datasets, mapping the complex molecular landscape of ASD [85]. Collectively, these studies illustrate how multi-omics analyses, combined with computational and imaging tools, can capture the breadth of molecular and functional alterations in ASD.

Single-cell and spatially resolved omics further enhance our understanding by mapping dysregulation to specific cell types and brain regions, which is critical in ASD and Rett syndrome, where excitatory neurons, interneurons, and glial populations are differentially affected [31,86]. Integration of single-cell multimodal omics with spatial transcriptomics has been shown to reveal the complex cellular architecture and intercellular communication in the brain, further enabling the identification of cell-type-specific disease signatures and molecular interactions critical for neurodevelopmental disorders [87]. Such spatially resolved approaches complement bulk and single-cell data by preserving tissue context, which is essential for deciphering region-specific pathophysiology and guiding targeted interventions.

Coupling single-cell transcriptomics with spatial proteomics or metabolomics can uncover region-specific regulatory networks and post-transcriptional modifications that bulk analyses may obscure [88]. Functional validation of multi-omics findings using patient-derived models, such as induced pluripotent stem cells (iPSCs) and cerebral organoids, further illuminates convergent molecular phenotypes and allows testing of candidate therapeutic interventions [89,90,91]. Similarly, computational models leveraging machine learning and network-based methods can predict functional impacts of variants and prioritize targets for experimental validation [92]. Worthy of note, the network-based approach is progressing from single-layer network representations (taking into account single omics data at a time) to multilayer network architectures that integrate heterogeneous omics data types. This new framework enables a more comprehensive view of perturbed complex systems, particularly relevant for understanding the multifactorial nature of complex disorder such as NDDs [93].

Longitudinal and multi-modal data integration will be essential for capturing the dynamic nature of neurodevelopment. Static snapshots may miss critical temporal alterations in gene expression, protein abundance, or metabolite flux, whereas repeated sampling combined with integrative multi-omics can illuminate disease trajectories, identify early biomarkers, and inform precision medicine strategies [1,7,80]. Translating these discoveries into clinical practice requires standardized data sharing, rigorous validation across cohorts, and integration with electronic health records and imaging data. Regulatory frameworks for multi-omics-based diagnostics and therapeutics remain nascent, emphasizing the need for reproducibility, transparency, and robust computational pipelines. As the field advances, these strategies hold promises to bridge the gap from molecular discovery to clinically relevant precision interventions in NDDs, ultimately improving diagnosis, prognosis, and therapeutic outcomes.

## 6. Conclusions

Integrative multi-omics approaches are reshaping our understanding of neurodevelopmental disorders (NDDs), offering a systems-level view of the molecular alterations underlying conditions such as ASD, ADHD, and ID. By combining omics data, these strategies have revealed convergent pathways, including synaptic dysfunction, immune dysregulation, and mitochondrial impairment, despite extensive genetic heterogeneity. 

Advanced computational methods, such as DIABLO, MOFA, and network-based models, are enabling the integration of high-dimensional, heterogeneous datasets, supporting molecular stratification and biomarker discovery. Notably, single-cell and spatially resolved multi-omics are allowing precise mapping of dysregulation to specific cell types and brain regions, as demonstrated in studies using cerebral organoids and scRNA-seq. Moreover, longitudinal designs and patient-derived models, including iPSCs, are beginning to capture dynamic disease trajectories and validate molecular findings in functional systems.

Despite remaining challenges, such as data harmonization, small cohort sizes, and the need for reproducibility, these integrative strategies are advancing the field toward clinically relevant applications. The convergence of multi-omics data with machine learning and imaging holds promise for precision diagnostics and targeted interventions, ultimately paving the way for personalized medicine in NDDs.

## Figures and Tables

**Figure 1 biomolecules-15-01401-f001:**
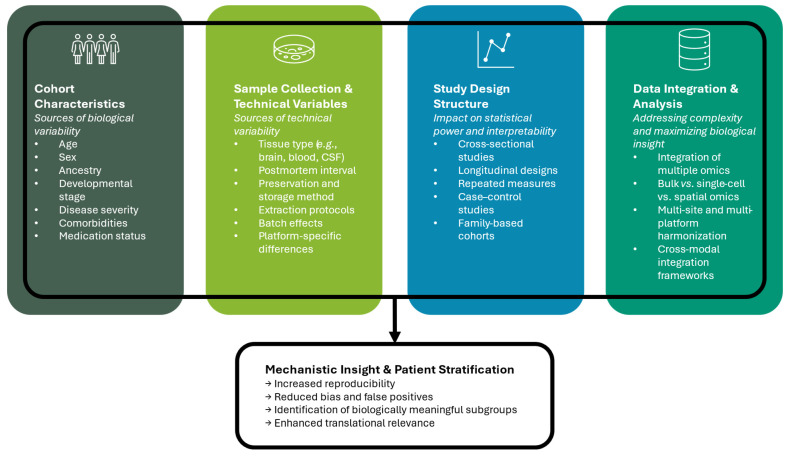
Overview of key factors influencing experimental design in multi-omics studies of neurodevelopmental disorders (NDDs). The figure illustrates major sources of variability that must be considered in study design, including biological characteristics of the cohort, sample-related and technical factors, and the type of study design employed. These elements contribute to both inter- and intra-subject heterogeneity and have critical implications for data quality, interpretability, and downstream analysis in multi-omics research.

**Table 1 biomolecules-15-01401-t001:** Key statistical and integrative methods in multi-omics analysis of neurodevelopmental disorders.

Method	Omics Layer	Strengths	Limitations	Refs.
DESeq2	Transcriptomics	Shrinkage of fold changes; robust FDR control	Needs replicates; univariate only	[19]
edgeR	Transcriptomics	Negative binomial modeling; small sample sizes	Sensitive to outliers	[20,28]
Limma	Transcriptomics/Proteomics	Linear modeling; empirical Bayes	Assumes log-normality	[28]
sPLS-DA	Multi-omics	Feature selection; dimensionality reduction	Sensitive to tuning; possible overfitting	[56]
sCCA	Multi-omics	Captures cross-dataset correlations	Needs regularization; computationally intensive	[47]
ComBat	Proteomics/Transcriptomics	Corrects batch effects	Risk of overcorrection	[24,25]
DIABLO	Multi-omics	Links features to phenotype	Requires normalized input; tuning complexity	[65]
SNF	Multi-omics	Patient stratification	Needs large cohorts; computational load	[66]
MOFA	Multi-omics	Handles missing data; interpretable latent factors	Hyperparameter sensitivity	[67]
Elastic Net/Priority-Elastic Net	Multi-omics	Feature selection; mitigates overfitting	Limited interpretability with correlated features	[41,60]

Note: columns show the method, the omics layer it applies to, main strengths, main limitations, and representative references.

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
