# Peer review of "Statistical Methods for Multi-Omics Analysis in Neurodevelopmental Disorders: From High Dimensionality to Mechanistic Insight"

_biomolecules, 2025, doi:10.3390/biom15101401_

Round 1

Reviewer 1 Report

Comments and Suggestions for Authors

The manuscript by Airoldi et a, provides an overview of statistical frameworks used to analyze high-dimensional omics data in the context of neurodevelopmental disorders (NDDs). The review is well-structured and covers important topics, including the statistical challenges when analyzing omics data, the distinction between univariate and multivariate models, and the importance of integrative multi-omics approaches. The review contains one figure and one table. Overall, the manuscript will be a valuable resource for researchers in the field. However, there are some areas that could be improved to enhance clarity, rigor, and the overall impact of the review. The following critique is provided to help strengthen the review and widen its impact.

Critique

  1. The review briefly touches on normalization and batch correction (lines 79-83) but omits a more thorough discussion of other critical preprocessing steps, such as quality control (QC) metrics, outlier removal, and the handling of confounding variables beyond a general mention in Figure 1. A more detailed account of how poor QC can undermine even the most sophisticated statistical analyses is a major missed opportunity.
  2. One area for improvement is in the section on imputation methods. While briefly discussed on line 177, the authors can significantly enhance their discussion of imputation methods by providing a more critical and detailed analysis of the available strategies. The current review briefly mentions methods like K-nearest neighbors (KNN) and random forest-based approaches but doesn't explain the trade-offs involved in choosing them. The revised section should articulate the practical implications of a researcher's choice, such as how a poor imputation strategy could affect the variance structure of the dataset. Furthermore, the manuscript should emphasize that while imputation can approximate missing measurements, the optimal solution is to prevent them during data collection, especially in proteomics, where observed zeros may reflect technical artifacts rather than true biological absence. This deeper level of detail would provide readers with the practical guidance needed to make informed decisions and ensure the robustness of their downstream analyses.
  3. A number of univariate analyses (e.g., DESeq2, edgeR, Limma, ComBat, SVA) are mentioned on lines 228-236 but their performance, applicability, or underlying assumptions in the context of NDDs is not discussed. For example, the choice between DESeq2 and edgeR often depends on specific data characteristics, and a discussion of these trade-offs would be highly valuable.
  4. The section on integrative multi-omics approaches (section 4, starting line 288) presents methods like DIABLO, SNF, and MOFA but fails to critically evaluate their limitations. The distinction between early integration (merging data before modeling) and late integration (combining results from independent analyses) is a useful way to categorize the methods. The authors provide excellent examples for early integration, such as DIABLO and SCCA. However, the concept of "late integration" (line 305) is described in terms of intersecting gene lists. This oversimplifies a diverse range of post-analysis strategies, such as pathway enrichment analysis or network-based integration. An expanded discussion of additional post-integration tools and methods will strengthen the manuscript.

Author Response

REVIEWER 1

The manuscript by Airoldi et al. provides an overview of statistical frameworks used to analyze high-dimensional omics data in the context of neurodevelopmental disorders (NDDs). The review is well-structured and covers important topics, including the statistical challenges when analyzing omics data, the distinction between univariate and multivariate models, and the importance of integrative multi-omics approaches. The review contains one figure and one table. Overall, the manuscript will be a valuable resource for researchers in the field. However, there are some areas that could be improved to enhance clarity, rigor, and the overall impact of the review. The following critique is provided to help strengthen the review and widen its impact.

Critique

  1. The review briefly touches on normalization and batch correction (lines 79-83) but omits a more thorough discussion of other critical preprocessing steps, such as quality control (QC) metrics, outlier removal, and the handling of confounding variables beyond a general mention in Figure 1. A more detailed account of how poor QC can undermine even the most sophisticated statistical analyses is a major missed opportunity.

Thank you for this thoughtful and constructive comment. We fully agree that rigorous preprocessing, including quality control (QC), outlier detection, and confounder adjustment, is essential to ensure the validity of downstream statistical analyses in multi-omics studies. In response, we have expanded the relevant section of the manuscript to more explicitly discuss the impact of QC and outlier removal, including how failures in these steps can propagate noise and bias in high-dimensional data. This revision strengthens the conceptual flow from data preprocessing to statistical modeling and integration.

  1. One area for improvement is in the section on imputation methods. While briefly discussed on line 177, the authors can significantly enhance their discussion of imputation methods by providing a more critical and detailed analysis of the available strategies. The current review briefly mentions methods like K-nearest neighbors (KNN) and random forest-based approaches but doesn't explain the trade-offs involved in choosing them. The revised section should articulate the practical implications of a researcher's choice, such as how a poor imputation strategy could affect the variance structure of the dataset. Furthermore, the manuscript should emphasize that while imputation can approximate missing measurements, the optimal solution is to prevent them during data collection, especially in proteomics, where observed zeros may reflect technical artifacts rather than true biological absence. This deeper level of detail would provide readers with the practical guidance needed to make informed decisions and ensure the robustness of their downstream analyses.

Thank you for this valuable suggestion. We agree that a more critical discussion of imputation methods would enhance the utility of the review. In response, we have expanded the relevant section to outline the strengths and limitations of commonly used imputation strategies, including their impact on variance structure and downstream analyses. We now also emphasize that poor imputation choices can introduce bias or distort biological signals, particularly in high-dimensional datasets such as proteomics. Moreover, we highlight that minimizing missingness at the experimental design stage remains the most robust strategy, especially for data-dependent acquisition (DDA) proteomics, where missing values often reflect stochastic sampling rather than true biological absence. These additions provide a more nuanced and practical guide for researchers navigating multi-omics imputation challenges.

  1. A number of univariate analyses (e.g., DESeq2, edgeR, Limma, ComBat, SVA) are mentioned on lines 228-236 but their performance, applicability, or underlying assumptions in the context of NDDs is not discussed. For example, the choice between DESeq2 and edgeR often depends on specific data characteristics, and a discussion of these trade-offs would be highly valuable.

Thank you for this insightful comment. We agree that a more detailed discussion of the methodological assumptions and comparative performance of DESeq2, edgeR, and Limma would enhance the practical value of the review. In response, we have added a short paragraph highlighting the key distinctions between these tools, focusing on their underlying models, suitability for different count distributions, and performance in small or heterogeneous cohorts typical of NDD studies. We also clarify the importance of selecting tools aligned with the specific characteristics of the dataset. Additionally, to balance the discussion between transcriptomics and proteomics, we have integrated recent examples of univariate statistical applications in proteomic studies of NDDs, including analyses using Limma and ANOVA in the context of biomarker discovery and developmental profiling. These additions aim to provide readers with a clearer rationale for tool selection across omics layers in neurodevelopmental research.

  1. The section on integrative multi-omics approaches (section 4, starting line 288) presents methods like DIABLO, SNF, and MOFA but fails to critically evaluate their limitations. The distinction between early integration (merging data before modeling) and late integration (combining results from independent analyses) is a useful way to categorize the methods. The authors provide excellent examples for early integration, such as DIABLO and SCCA. However, the concept of "late integration" (line 305) is described in terms of intersecting gene lists. This oversimplifies a diverse range of post-analysis strategies, such as pathway enrichment analysis or network-based integration. An expanded discussion of additional post-integration tools and methods will strengthen the manuscript.

Thank you for this insightful comment. We agree that the initial description of late integration as simply intersecting gene or protein lists was too narrow. We have expanded the manuscript to include additional post-analysis strategies such as pathway and gene set enrichment analyses, network-based integration approaches, and meta-dimensional methods that combine independent omics results while preserving layer-specific information. Moreover, we had added a concrete example of how late integration methods can lead to insightful results. However, to maintain focus and clarity in our review, which centers primarily on statistical methods, we have limited the depth of this discussion. We believe this balance strengthens the manuscript while keeping it aligned with its core focus.

Reviewer 2 Report

Comments and Suggestions for Authors

Journal: Biomolecules (ISSN 2218-273X)

Manuscript ID: biomolecules-3889655

Type: Review

Title: Statistical Methods for Multi-Omics Analysis in Neurodevelopmental Disorders: From High Dimensionality to Mechanistic Insight

Authors: Manuel Airoldi , Veronica Remori , Mauro Fasano *

Section: Bioinformatics and Systems Biology

This review is well-written, comprehensive, and timely, providing a clear overview of statistical and computational methods for integrating multiple omics in neurodevelopmental disorders. The manuscript is well-organized, informative, and useful to methodologists and neuroscientists alike. Minor revisions are required primarily for editorial polishing (typos, redundant phrasing, and abbreviation consistency) and to improve figure/table clarity. Addressing these minor issues will increase readability and impact, making the article a valuable contribution to the field.

1: Lines 54-55: “…underscoring the necessity of complementary molecular and computational strategies [11].”Dangling phrase.

it should be “…which underscores the necessity of complementary molecular and computational strategies [11].”

2: The abstract and discussion state that multi-omics approaches reveal mechanistic insights, but the manuscript does not provide concrete mechanistic case studies.

 Include explicit examples that demonstrate how multi-omics led to specific mechanisms.

3: Overemphasis on methods that lack biological integration (lines 103-221, 222-283). The statistical descriptions are detailed, but the links to neurodevelopmental biology are occasionally weak. Strengthen connections between methods and how they reveal biological characteristics of ASD, ID, or Rett syndrome.

4: Lines 297-307: Insufficient critical comparison of integration strategies.

 Early vs. late integration strategies are mentioned, but their relative advantages and disadvantages are not thoroughly examined.

Provide a critical evaluation (with examples) of when each strategy is most appropriate.

5: Abbreviations missing in the first mention (e.g., ICA)

Some abbreviations were introduced without definition.

Define on first appearance.

6: Mechanistic depth - The manuscript provides a good summary of the methods but does not go into detail about how multi-omics approaches lead to specific mechanistic insights in NDDs.

7: Validation and reproducibility - There is a limited discussion of independent dataset validation, benchmarking, and reproducibility issues.

Author Response

REVIEWER 2

This review is well-written, comprehensive, and timely, providing a clear overview of statistical and computational methods for integrating multiple omics in neurodevelopmental disorders. The manuscript is well-organized, informative, and useful to methodologists and neuroscientists alike. Minor revisions are required primarily for editorial polishing (typos, redundant phrasing, and abbreviation consistency) and to improve figure/table clarity. Addressing these minor issues will increase readability and impact, making the article a valuable contribution to the field. 

1: Lines 54-55: “…underscoring the necessity of complementary molecular and computational strategies [11].”Dangling phrase.

it should be “…which underscores the necessity of complementary molecular and computational strategies [11].”

Thank you for pointing this out. We have revised the sentence as suggested to improve clarity and readability.

2: The abstract and discussion state that multi-omics approaches reveal mechanistic insights, but the manuscript does not provide concrete mechanistic case studies.

 Include explicit examples that demonstrate how multi-omics led to specific mechanisms.

Thank you for this important feedback. In response to this comment, we now discuss two illustrative studies applying multi-omics data integration approaches in ASD and ADHD (section 4 of the manuscript). These examples highlight how such strategies can provide mechanistic insights into NDDs. We believe this enhancement strengthens the manuscript by providing concrete evidence

3: Overemphasis on methods that lack biological integration (lines 103-221, 222-283). The statistical descriptions are detailed, but the links to neurodevelopmental biology are occasionally weak. Strengthen connections between methods and how they reveal biological characteristics of ASD, ID, or Rett syndrome.

Thank you for this constructive comment. We have addressed it by adding examples of how these methods have been applied to NDDs. At the same time, we chose not to go into excessive detail, as the first three sections of the manuscript are intended to provide an introductory overview of the statistical approaches. We believe this balance improves the manuscript by highlighting biological applications while preserving its introductory focus.   

4: Lines 297-307: Insufficient critical comparison of integration strategies.

 Early vs. late integration strategies are mentioned, but their relative advantages and disadvantages are not thoroughly examined.

Provide a critical evaluation (with examples) of when each strategy is most appropriate.

Thank you for your valuable comment. We have expanded the section to include a concise discussion of the advantages and disadvantages of early and late integration strategies, highlighting when each approach is most suitable in the context of neurodevelopmental disorder studies.

5: Abbreviations missing in the first mention (e.g., ICA)

Some abbreviations were introduced without definition.

Define on first appearance.

Thank you for your comment. We have carefully reviewed all abbreviations in the manuscript and ensured that each one is defined upon its first appearance.

6: Mechanistic depth - The manuscript provides a good summary of the methods but does not go into detail about how multi-omics approaches lead to specific mechanistic insights in NDDs.

Thank you for this feedback. We have addressed it by including examples of studies in which multi-omics approaches have led to novel mechanistic insights in neurodevelopmental disorders such as ADHD and ASD. These addition help illustrate how the integration of multi-omics data can reveal biological mechanisms underlying these complex conditions

7: Validation and reproducibility - There is a limited discussion of independent dataset validation, benchmarking, and reproducibility issues.

We thank the reviewer for the valuable feedback. We added a paragraph at the end of section 2 to emphasize the importance of validation, benchmarking, and reproducibility in multi-omics studies.